# Cytomegalovirus-Driven Adaption of Natural Killer Cells in NKG2C^null^ Human Immunodeficiency Virus-Infected Individuals

**DOI:** 10.3390/v11030239

**Published:** 2019-03-09

**Authors:** Emilie M. Comeau, Kayla A. Holder, Neva J. Fudge, Michael D. Grant

**Affiliations:** Immunology and Infectious Diseases Program, Division of BioMedical Sciences, Faculty of Medicine, Memorial University of Newfoundland, 300 Prince Philp Drive, St. John’s, NL A1B 3V6, Canada; emc878@mun.ca (E.M.C.); lkaylaaholder@gmail.com (K.A.H.); nfudge@mun.ca (N.J.F.)

**Keywords:** HCMV, HIV, NKG2C, NKG2C^null^, NKG2A, CD16, CD2, PLZF, FcεRIγ, ADCC

## Abstract

Expansion of natural killer (NK) cells expressing NKG2C occurs following human cytomegalovirus (HCMV) infection and is amplified by human immunodeficiency virus (HIV) co-infection. These NKG2C-expressing NK cells demonstrate enhanced CD16-dependent cytokine production and downregulate FcεRIγ and promyelocytic leukemia zinc finger protein (PLZF). Lacking NKG2C diminishes resistance to HIV infection, but whether this affects NK cell acquisition of superior antibody-dependent function is unclear. Therefore, our objective was to investigate whether HCMV-driven NK cell differentiation is impaired in NKG2C^null^ HIV-infected individuals. Phenotypic (CD2, CD16, CD57, NKG2A, FcεRIγ, and PLZF expression) and functional (cytokine induction and cytotoxicity) properties were compared between HIV–infected NKG2C^null^ and NKG2C-expressing groups. Cytokine production was compared following stimulation through natural cytotoxicity receptors or through CD16. Cytotoxicity was measured by anti-CD16-redirected lysis and by classical antibody-dependent cell-mediated cytotoxicity (ADCC) against anti-class I human leukocyte antigen (HLA) antibody-coated cells. Our data indicate highly similar HCMV-driven NK cell differentiation in HIV infection with or without NKG2C. While the fraction of mature (CD57^pos^) NK cells expressing CD2 (*p* = 0.009) or co-expressing CD2 and CD16 (*p* = 0.03) was significantly higher in NKG2C^null^ HIV-infected individuals, there were no significant differences in NKG2A, FcεRIγ, or PLZF expression. The general phenotypic and functional equivalency observed suggests NKG2C-independent routes of HCMV-driven NK cell differentiation, which may involve increased CD2 expression.

## 1. Introduction

Human cytomegalovirus (HCMV) is a large double-stranded DNA beta-herpes virus that has established lifelong, primarily latent infection in a large fraction of the global population [1,2]. In healthy hosts, reactivation of HCMV is generally well controlled by a robust immune response and, therefore, remains clinically asymptomatic [1,2,3]. However, over time, both the adaptive and innate branches of the human immune system evolve towards dedicating an inordinate fraction of their resources towards HCMV surveillance [4]. Infection with HCMV triggers expansion of a usually minor CD57^pos^NKG2C^pos^ natural killer (NK) cell subset and expansion of this subset is exaggerated among immunocompromised hosts, such as transplant recipients and individuals infected with human immunodeficiency virus (HIV) [5,6,7,8,9,10,11,12,13]. An increased proportion of peripheral blood NK cells expressing the activating receptor NKG2C clearly illustrates the selective influence of HCMV infection imprinted on the host immune system. These NKG2C^pos^ NK cells usually co-express the maturation marker CD57 and are functionally distinguished from the rest of the NK cell population by enhanced CD16-dependent cytokine production [14,15]. The CD57^pos^NKG2C^pos^ NK cells also undergo epigenetic remodeling of the interferon (IFN)-γ promoter region to favor increased production of IFN-γ [16,17,18,19,20,21]. Since their expression is, for the most part, mutually exclusive, another notable feature of the expanded NK cell population is absence of NKG2A, which is the inhibitory counterpart to NKG2C [22]. Further phenotypic and functional adaptation of NK cells to HCMV infection is reflected in reduced levels of the promyeloctic leukemia zinc finger protein (PLZF) transcription factor and reduced levels of FcεRIγ, which is a signaling adaptor protein associated with CD16 [18,19,21]. Reduced expression of this adaptor protein is actually associated with enhanced CD16-dependent signaling, potentially through replacement of FcεRIγ with a more potent adaptor protein, such as CD3ζ [23]. Despite conventional depiction of NK cells as innate immune cells, a large proportion of CD57^pos^NKG2C^pos^ NK cells acquire phenotypic and functional features associated with adaptive immune memory in response to HCMV infection. Maturation of NK cells into memory-like cells with enhanced cytokine responses to stimulation through CD16 has been reported to occur similarly in HIV-infected individuals infected with HCMV [24].

Although initially associated with NKG2C expression, memory NK cell features are not restricted to NKG2C^pos^ NK cells. Homozygous deletion of a 16 kb region of the genome encompassing the NKG2C gene (KLRC2) is relatively common [25]. Individuals with this deletion, herein termed NKG2C^null^, generally maintain subclinical HCMV infection, even in the case of co-infection with HIV, but are at greater risk for more rapid disease progression once HIV infection occurs [26,27,28]. Maturation of NK cells in response to HCMV infection does take place for NKG2C^null^ individuals, but the rate and extent of maturation compared to that of NK cells expressing NKG2C, and whether compensatory features are involved, remain in question [5,29,30,31,32]. Thus, there is no consensus on whether NK cell maturation in response to HCMV infection is impaired in NKG2C^null^ hosts and little has been reported on NK cell phenotype and function in NKG2C^null^ HCMV-infected individuals co-infected with HIV. Since HIV-infected individuals infected with HCMV generally have larger populations of CD57^pos^NKG2C^pos^ differentiated NK cells than do healthy controls infected with HCMV, we reasoned that any deficits or detours in NK cell differentiation related to a lack of NKG2C expression might be more readily visualized in this setting [5,13]. Therefore, we investigated the effect of NKG2C gene deletion on phenotypic and functional NK cell maturation in HIV-infection by analyzing NK cells from NKG2C^null^ and NKG2C-expressing hosts matched for age and HCMV/HIV infection history. Our aim was to determine whether HCMV-driven NK cell maturation into enhanced effectors is comparable between NKG2C^null^ and NKG2C-expressing individuals.

## 2. Materials and Methods

### 2.1. Sample Collection and Peripheral Blood Mononuclear Cell Isolation

This study received approval from the Health Research Ethics Authority of Newfoundland and Labrador, Canada. Study subjects were recruited through the Newfoundland and Labrador Provincial HIV Clinic in St. John’s, Newfoundland and gave written informed consent and access to clinical laboratory information in accordance with the Declaration of Helsinki. This study was carried out in accordance with the recommendations of the Canadian Tri-Council Policy Statement: Ethical Conduct for Research Involving Humans. Blood was collected by forearm venipuncture into acid-citrate dextrose-containing vacutainers and peripheral blood mononuclear cells (PBMC) were isolated by density gradient centrifugation using Ficoll-Paque PLUS lymphocyte isolation medium (GE Healthcare, Chicago, IL, USA). Cells were counted, resuspended in freezing medium at 1 × 10^7^/mL, cooled to −80 °C at 1 °C/minute overnight, and stored in liquid nitrogen until use. Freezing medium was lymphocyte medium: RPMI with 10% fetal calf serum (FCS), 200 IU/mL penicillin/streptomycin (P/S), 1% 1 M HEPES, 1% L-glutamine (all from Invitrogen, Carlsbad, CA, USA), and 2.0 × 10^−5^ M 2-mercaptoethanol (Sigma-Aldrich, St. Louis, MO, USA) with 10% dimethyl sulfoxide and supplemented to 20% FCS. Samples were incubated overnight at 37 °C in 5% CO_2_ in lymphocyte medium for recovery.

### 2.2. Identification of NKG2C^null^ Individuals and a Matched Control Group

Individuals with <1% NKG2C-expressing NK cells (CD3^neg^CD56^pos^) by flow cytometry were subsequently genotyped by polymerase chain reaction (PCR) to test for homozygous KLRC2 deletion. Amplification involved two sets of forward and reverse primers, “NKG2C” and “Break” (Integrated DNA Technologies, Coralville, IA, USA), designed by Miyashita et al. to detect NKG2C gene deletions [25]. The PCR conditions established in our laboratory differed from those of Miyashita et al. in the annealing conditions (30 s at 45 °C for annealing), and that both NKG2C and Break reactions were amplified under identical conditions. After gel electrophoresis, DNA bands were visualized with SYBR Safe™ DNA gel stain (Invitrogen, Carlsbad, CA, USA) using the Kodak Gel Logic 440 Imager (Kodak, Rochester, NY, USA).

Eight NKG2C^null^ individuals identified by PCR genotyping in our study cohort of HIV-infected individuals were matched to an HIV-infected NKG2C-expressing control group (NKG2C^match^) based primarily on age, duration of HIV infection, and CD4^pos^ T cell nadir. All NKG2C^null^ individuals and matched controls were co-infected with HCMV. We also compared anti-HCMV IgG levels measured by enzyme-linked immunosorbent assay and anti-CMV CD8^pos^ T cell responses, which are measured as IFN-γ-expressing CD8^pos^ T cells post-stimulation with overlapping HCMV peptides from immunodominant pp65 and IE-1 proteins between the groups [5,33] (see Table 1 for NKG2C^null^ and NKG2C^match^ subject characteristics).

### 2.3. Flow Cytometry

PBMC were stained for extracellular markers (clones in parenthesis) with anti-CD2 (RPA-2.10, BioLegend, San Diego, CA, USA), anti-CD3 (BW264/56), anti-CD16 (3G8), anti-CD56 (REA196), anti-CD57 (TB03), anti-NKG2A (REA110), all from Miltenyi Biotec, Auburn, CA, USA and anti-NKG2C (134591, R&D Systems, Minneapolis, MN, USA). LIVE/DEAD Fixable Far Red (Invitrogen) was used to exclude dead cells. Fixation and permeabilization were performed using MACS Inside Stain Kit (Miltenyi Biotec) with polyclonal goat anti-human FcεRIγ (EMD Millipore, Etobicoke, ON, Canada), anti-IFN-γ (4S.B3), and anti-TNF-α (MAb11) from eBioscience, San Diego, CA, USA added for intracellular staining. The fluorescence minus one strategy was used to adjust multicolor compensation. Phenotypic analysis was performed using Kaluza Flow Cytometry Software 1.2 after data acquisition on a MoFlo Astrios EQ flow cytometer (both Beckman Coulter, Brea, CA, USA).

### 2.4. NK Cell Stimulations

To measure general NK cell activation in vitro through natural cytotoxicity receptors (NCR), class I human histocompatibility-linked antigen (HLA) deficient erythroleukemia K562 cells (ATCC^®^ #CCL 243™) were incubated with PBMC at an effector to target (E:T) ratio of 5:1 (2 × 10^6^ PBMC:4 × 10^5^ K562 cells). To selectively measure antibody-dependent NK cell activation in vitro, 2 µg anti-CD16 monoclonal antibody (LEAF™ 3G8, BioLegend), was added to 2 × 10^6^ PBMC in 2 mL lymphocyte medium. After one hour, Brefeldin A (Sigma-Aldrich) was added to both stimulations to a final concentration of 10 µg/mL followed by an additional 15 h incubation whereby IFN-γ and TNF-α production was measured after intracellular staining. All cell lines used were cultured at 37 °C in 5% CO_2_ in lymphocyte medium and maintained in a log phase growth for labeling.

### 2.5. Cytotoxicity Assays

Two NK cell cytotoxicity assays were used to compare CD16-mediated NK cell cytotoxicity between groups. The C1R-B27 B lymphoblastoid cell line (Dr. Kelly MacDonald, University of Manitoba, Canada) coated with anti-HLA class I antibodies served as the NK cell target in a classical ADCC assay [34]. The pan-anti-HLA class I antibody (Ab) used in this assay at 1 μg/mL was produced by the W6/32 murine B cell hybridoma (ATCC^®^ HB-95™). The second assay measured killing of the murine mastocytoma cell line, P815 (ATCC^®^ TIB-64™) by anti-CD16 redirected lysis using LEAF™ 3G8. Approximately 1 × 10^6^ target cells were pelleted and labeled with 100 µCi sodium ^51^chromate (Na_2_^51^CrO_4_, Perkin-Elmer, Waltham, MA, USA) for 90 min. Cytotoxicity assays were performed in duplicate in U-bottom 96-well microtiter plates (BD Biosciences, San Jose, CA, USA) at E:T 50:1. Serial dilutions of anti-CD16 and anti-HLA class I antibodies estimated half-maximal effective concentrations (EC_50_). P815 cells were plated with 12 doubling dilutions of LEAF™ 3G8 from an initial concentration of 100 ng/mL. C1R-B27 cells were incubated with six doubling dilutions of the W6/32 antibody, starting from 1 μg/mL. Test wells contained 200 µL diluted Ab in lymphocyte medium, 50 µL ^51^Cr-labeled target cells at an initial concentration of 1 × 10^5^/mL, and 50 µL PBMC at an initial concentration of 5 × 10^6^/mL. Maximum and minimum release wells contained 50 μL targets with 250 μL 1N hydrochloric acid or lymphocyte medium, respectively. Both assays incorporated no-Ab controls as the background. After 5 h of incubation, 110 µL of sample was removed from each well and the ^51^Cr released was counted on a Wallac 1480 Wizard Gamma Counter (Perkin-Elmer). Percent specific lysis was calculated as [(experimental ^51^Cr release − minimum ^51^Cr release)/(maximum ^51^Cr release − minimum ^51^Cr release)] × 100.

### 2.6. Statistical Analysis

Statistical analysis was performed using GraphPad Prism software version 6 (GraphPad Software, La Jolla, CA, USA). Data were represented as mean ± standard deviation (SD) when normally distributed or as median ± interquartile range (IQR) when not normally distributed. Normal distribution was assessed by D’Agostino and Pearson and Shapiro-Wilk normality tests. If either test indicated non-normal distribution, the data were treated as not normally distributed. If data were normally distributed, the unpaired Student’s *t* test was used to assess the probability of intergroup differences, and the Mann-Whitney *U* test when data were not normally distributed. Comparisons within groups were carried out with paired Student’s *t* test when data were normally distributed and with the Wilcoxon signed rank test otherwise. Differences were considered statistically significant when the two tailed p value was ≤0.05.

## 3. Results

### 3.1. General Characteristics of NKG2C^null^ and NKG2C^match^ Groups

An HIV-infected control group was selected for comparison with the eight NKG2C^null^ HIV-infected individuals we identified based on age, sex, and previous extent of the HIV-related disease progression (CD4^pos^ T cell nadir). All subjects in both groups were receiving a combination of anti-retroviral therapy (cART) with a duration on treatment ranging from 5 to 23 years in the NKG2C^null^ group and from 4 to 23 years in those selected as matches. Occasional blips or viral breakthroughs occurred over the duration of infection, but, for the most part, a virus load was maintained below detectable levels. All subjects were seropositive for HCMV with no significant difference in the percentage of NK cells in the lymphocyte population or in anti-CMV IgG levels, as measured by ELISA. However, the NKG2C^null^ group had a significantly greater CD8^pos^ T cell response (*p* = 0.05, Mann-Whitney *U* test) against the immunodominant pp65 and IE-1 HCMV antigens, compared to the matched controls (Table 1).

### 3.2. Phenotypic Comparison of NK Cells from NKG2C^null^ and a Matched Control Group

We compared phenotypic NK cell features between groups by flow cytometry with gating, as shown (Figure 1A–F). LIVE/DEAD stain was used to exclude dead cells after gating on the lymphocyte population (Figure 1B) and CD3^neg^CD56^pos^ lymphocytes in the lower right quadrant (Figure 1C) were selected for further phenotypic analysis. We then compared expression levels of CD2, CD16, and CD57, which are general markers for NK cell maturation (Figure 1G–I), and the expression of NKG2A, FcεRIγ, and PLZF, the loss of which relate to HCMV-driven NK cell maturation (Figure 1J–L). No significant differences were observed in comparing mean or median levels of any of these markers between the NKG2C^null^ and matched control groups. This indicates that NK cells from NKG2C^null^ HIV-infected individuals undergo similar phenotypic maturation to those from HIV-infected individuals expressing KLRC2.

To investigate whether NK cell maturation followed a similar pattern in the NKG2C^null^ and NKG2C^match^ groups, we compared levels of the same markers on the entire NK cell population versus the mature (CD57^pos^) NK cell populations within the two groups and on mature NK cell populations between the two groups. For CD2, a significantly higher percentage of mature NK cells expressed this marker in the NKG2C^null^ group (*p* = 0.036), while there was no significant difference in the NKG2C^match^ group (Figure 2A). A higher percentage of mature NK cells expressed CD16 in both groups (*p* = 0.031, Figure 2B) or co-expressed CD2 and CD16 (*p* = 0.031, Figure 2C). Despite levels of these markers not being significantly different between groups in terms of the general NK cell population (Figure 1), a significantly higher percentage of mature NK cells within the NKG2C^null^ group expressed CD2 (*p* = 0.009, Figure 2A), or co-expressed CD2 and CD16 (*p* = 0.03, Figure 2C) when compared to the NKG2C^match^ group. A significantly lower percentage of mature NK cells expressed the NKG2A inhibitory receptor compared to the reciprocal population in both groups (*p* = 0.031, Figure 2D). These results suggest that there may be selective expansion and maturation of CD2-expressing NK cells in HIV-infected individuals lacking the NKG2C gene.

Loss of FcεRIγ and PLZF from NK cells is associated with more potent antibody-dependent cytokine production in HCMV-infected individuals and is reportedly more common among the mature (CD57^pos^) NK cell population [8,11,21]. Our results indicated a similar mean proportion of mature CD57^pos^ NK cells in both groups (Figure 1F) and, in contrast with previous reports, we found no selective increase in the FcεRIγ^neg^ NK cell fraction among mature CD57^pos^ NK cell populations compared to the entire NK cell population (Figure 2E). The percentage of NK cells expressing PLZF also did not differ significantly between groups, but, as we observed for NKG2A, the number of mature NK cells lacking expression of PLZF was significantly increased compared to the reciprocal population in both the NKG2C^null^ (*p* = 0.0007) and NKG2C^match^ (*p* = 0.0025) groups (Figure 2F). Thus, NK cells from both of these groups infected with HIV behave similarly in terms of losing expression of NKG2A, FcεRIγ, and PLZF in adaptation to HCMV infection.

### 3.3. Comparison of Cytokine Production by NK Cells from NKG2C^null^ and NKG2C^match^ Groups

To investigate whether the general phenotypic equivalence of the groups’ NK cells extended across their function, we first measured the fractions of NK cells expressing IFN-γ or TNF-α following stimulation with K562 cells. Similar fractions of NK cells from both groups responded to stimulation with K562, measured by IFN-γ and TNF-α expression (Figure 3A and Figure 3B, respectively). These data indicate that NK cells from NKG2C^null^ and NKG2C^match^ groups were functionally equal in terms of general activation with response to stimulation through NCR by K562 cells. Since adapted NK cells are reportedly superior in terms of antibody-dependent cytokine production, we next measured NK cell activation through CD16 receptors [14,15]. Responses triggered by anti-CD16 (Figure 3C,D) were notably stronger than those stimulated by K562 cells (Figure 3A,B). Again, we saw no significant differences between groups in either IFN-γ or TNF-α production in response to stimulation through CD16 (Figure 3C and Figure 3D, respectively). Our results also indicate equivalence between groups in terms of pro-inflammatory cytokine responses to CD16 stimulation, as measured by IFN-γ and TNF-α expression. These results are generally consistent with functional equivalency of NK cells from NKG2C^null^ and matched controls co-infected with HCMV and HIV in terms of pro-inflammatory cytokine production following stimulation though either NCR or through CD16.

### 3.4. Comparison of ADCC by NK Cells from the NKG2C^null^ and Matched Control Groups

We next compared NK cell cytotoxicity against antibody-treated cell lines in anti-CD16 redirected cytotoxicity and classical ADCC assays. At optimal Ab concentrations, there was no significant difference between the NKG2C^null^ and NKG2C^match^ groups in the level of either anti-CD16 redirected lysis of P815 cells (Figure 4A) or antibody-dependent lysis of C1R-B27 cells (Figure 4B). These data indicate equivalence between groups in their capacity for ADCC and anti-CD16 redirected lysis regardless of NKG2C expression. However, measuring cytotoxicity at only one Ab concentration does not address possible differences in sensitivity to triggering through CD16. To test whether the sensitivity of triggering through CD16 differed between groups, we repeated cytotoxicity assays with serial dilutions of 3G8, which ranged over 12 doubling dilutions from 100 ng/mL to 0.05 ng/mL, and anti-HLA-I Ab W6/32, which ranged from 1000 ng/mL to 16 ng/mL over six doubling dilutions. Figure 4C,D represent Ab titrations. The 50% effective Ab concentration (EC_50_), which is defined as the concentration required to elicit a half maximal cytotoxicity response, was estimated for NK cells by each subject and compared between groups. Both the NKG2C^null^ and NKG2C^match^ groups had a similar mean EC_50_ of ~125 ng/mL for W6/32 Ab titration in ADCC assays and were similar with respect to the 3G8 titration for redirected P815 cell lysis at ~1.2 ng/mL. The highly similar EC_50_ values for both groups suggest comparable sensitivity to triggering through CD16 between NK cells from NKG2C^null^ and NKG2C^match^, HIV-infected individuals co-infected with HCMV. In order to visually convey the functional similarity across the groups, despite the high variability in the absolute levels of killing, we normalized our results to 100% to show largely overlapping titration curves from both the NKG2C^null^ and NKG2C^match^ groups (Figure 4E,F).

## 4. Discussion

The activating receptor NKG2C is expressed on a population of mainly mature, functionally adapted NK cells that expands following HCMV infection [35]. Selective recognition of peptides from HCMV by NKG2C in the context of HLA-E driven NK cell proliferation implies a direct role for NKG2C in NK cell adaptation and functional differentiation to HCMV infection [36,37,38,39,40,41]. However, elements of NK cell adaptation to HCMV also occur independently of NKG2C [29,30,31,42]. As expansion of CD57^pos^NKG2C^pos^ NK cells is especially pronounced in HIV-infected individuals co-infected with HCMV, we reasoned that any deficits in NK cell phenotypic or functional maturation imposed by NKG2C deficiency would be prominent in this setting [5,13]. A previous study found no accelerated or exaggerated expansion of CD57^pos^NKG2C^pos^ NK cells in HCMV-infected individuals co-infected with hepatitis B virus (HBV), HCV, or HDV [43]. Therefore, we identified individuals in our HIV-infected study cohort homozygous for NKG2C gene deletion and compared relevant attributes of their NK cells with those of HIV-infected controls matched for age, sex, and HIV infection history. In general, we observed little difference between total NK cells of the two groups either functionally or phenotypically. There was no significant difference in the overall fraction of total NK cells expressing CD2, CD16, CD57, NKG2A, FcεRIγ, or PLZF, which indicates similar levels of general maturation and adaptation to HCMV infection.

Both groups had fewer PLZF or NKG2A-expressing and more CD16-expressing NK cells within their mature CD57^pos^ populations, which illustrates a similar maturation pattern irrespective of NKG2C expression. Neither group showed an increased prevalence of NK cells lacking FcεRIγ in the mature population, but we did note a selective increase in CD2 expression and its co-expression with CD16 on mature CD57^pos^ NK cells of the NKG2C^null^ group. This finding corroborates several previous studies on HIV-naïve NKG2C^null^ subjects from which suggestions of an enhanced role for CD2 in triggering their NK cells arose, especially in tandem with CD16 stimulation [31,44]. Our functional studies detected no significant differences between the two groups in terms of IFN-γ or TNF-α production following stimulation through NCRs with K562 cells or stimulation through CD16 with mAb 3G8. Absolute levels of cytotoxicity mediated through CD16 either in a conventional ADCC assay or by redirected lysis, which were also not significantly different between groups. Sensitivity to activation through CD16 was equivalent. While we did not measure direct killing of K562 cells, the ability of the NK cells to produce multiple cytokines and to kill target cells through ADCC indicates similar retention or acquisition of polyfunctionality.

These data indicate that, in the setting of HIV infection, any contribution NKG2C makes in NK cell adaptation to HCMV infection is effectively compensated for in NKG2C^null^ individuals. One potential component of this compensation is increased expression of CD2 and a greater role for it as a co-stimulatory receptor in CD16-mediated NK cell activation. In a recent study, adapted NK cells from NKG2C-sufficient and NKG2C-deficient subjects were deemed phenotypically, functionally, and epigenetically similar, with only NKG2C expression itself being a distinguishing factor [31]. In this same study, CD2 and CD16 co-ligation increased NK cell cytokine production, which supported the suggestion that CD2 may play a greater co-stimulatory role in this setting. However, hypomethylation of the CD2 locus is a general feature of adapted NK cells and CD2 is, at least somewhat, upregulated on adapted NK cells from NKG2C-sufficient and NKG2C-deficient subjects alike [20,31]. Thus, while CD2 may be one mechanism whereby NKG2C-deficient individuals compensate with respect to NK cell function, other factors could also include increased prevalence of adapted NK cells expressing activating killer cell immunoglobulin-like receptors [45]. While the lack of significant difference in NK cell phenotype and function of NKG2C^null^ compared to NKG2C-expressing individuals may be one more example of immune system redundancy, the ultimate significance of an expanded population of NKG2C^pos^ NK cells and role of NKG2C towards controlling HCMV infection or stimulating NK cell expansion and adaptation might be questioned.

A variable role for NKG2C in NK cell function could also reflect the nature of NK cell regulation, in which multiple positive and negative signals are integrated to determine the outcome of cellular interactions. In the case of NKG2C sufficiency, interaction with HLA-E provides a significant activating signal, which incorporates a degree of antigenic specificity through the propensity of certain HCMV-derived peptides to bind HLA-E and increase its surface expression [37,38,39,40]. Despite this, no direct role in controlling HCMV replication has been shown and the overall adaptive value of this pathway remains unclear. Upregulation of CD2 on NK cells may be a generic response to HCMV infection since the HCMV UL148 gene product reduces expression of CD58, which restricts co-stimulation through CD2 and reduces HCMV-specific T cell and NK cell activation [46]. Enhanced signaling through CD2, which was demonstrated for NK cells from NKG2C^null^ persons, could compensate generally for NKG2C deficiency, but might be of little value against HCMV specifically [31,44].

Although it is HCMV and not HIV driving the maturation and expansion of NKG2C^pos^ NK cells, susceptibility to HIV infection and disease progression is associated with the NKG2C^null^ genotype [26,27,28]. While there appears to be a compensatory mechanism for NK cells to address NKG2C deficiency, this receptor is also expressed on a subset of T cells and reportedly contributes to γδ T cell recognition of HIV-infected cells [47]. Thus, the HIV sensitivity attributed to lack of NKG2C may not completely reflect an impact on NK cell function. There is a single case report of a T cell deficient child partially controlling HCMV infection in association with NKG2C^pos^ NK cell expansion, but this may not be a critical aspect of immunity to HCMV in persons with an intact immune system [8]. Although the T cell responses we measured were limited to two immunodominant HCMV proteins, we did observe a significantly greater median HCMV-specific T cell response in the NKG2C^null^ group, which is consistent with the idea of HCMV infection placing greater demand on adaptive immunity to control HCMV in the setting of NKG2C deficiency. A significant correlation was reported in HIV-infected persons between the fraction of NK cells expressing NKG2C and the fraction of HCMV-specific CD8^pos^ T cells having lost expression of CD28, which could be interpreted as both arms of the HCMV-related immune response being stressed to control HCMV replication in HIV-infected individuals [5]. Amplification of the adaptive cellular immune response against HCMV in HIV-infected individuals to levels normally seen in people without HIV infection three or four decades older has been likened to premature aging of the immune system [48]. The similar amplification of NK cell responses to HCMV infection in the setting of HIV infection may also reflect accelerated immune aging [5,15].

A key feature of HCMV-adapted NK cells that investigators wish to apply practically is their broad-based superiority in mediating CD16-dependent effector functions [14,18,19]. This feature could be beneficial against any microbe or transformed cell susceptible to these effector functions. While it appears that NKG2C expression plays a role in the expansion or adaptation of these cells in certain circumstances, it may be disposable for the process. As our NKG2C^null^ donors appeared at no functional disadvantage compared to matched controls, it appears that NK cells from NKG2C^null^ donors have alternative routes of differentiation, which involve or culminate in synergistic activity between CD2 and CD16. Understanding alternate, potentially complementary routes of HCMV-driven NK cell differentiation may inform strategies to more effectively harness NK cells therapeutically against infections and cancer.

## Figures and Tables

**Figure 1 viruses-11-00239-f001:**
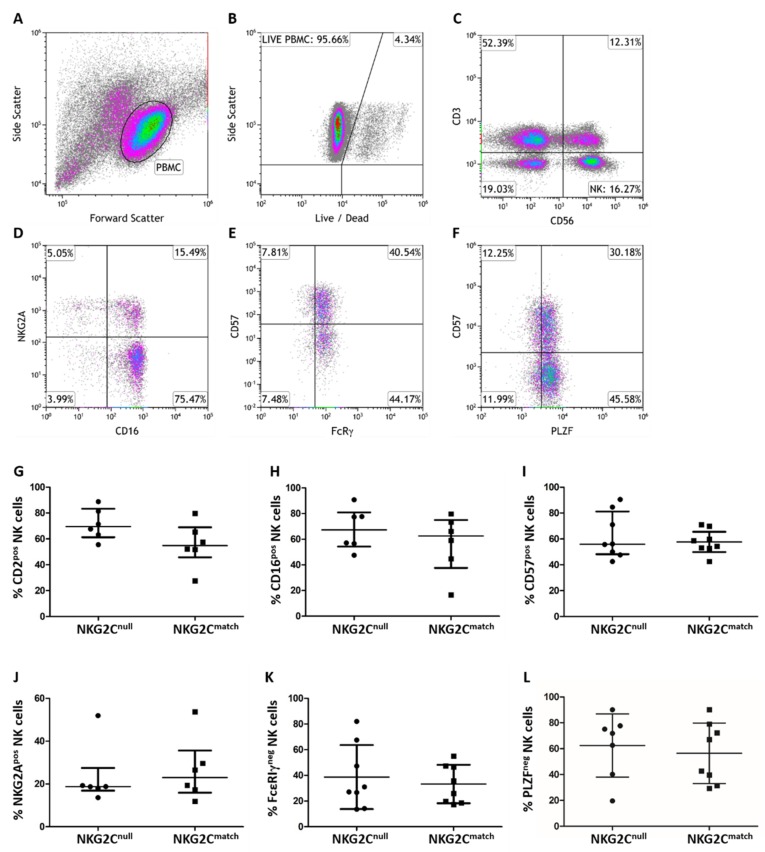
Phenotypic comparison of NK cells from NKG2C^null^ and matched subjects. Cryopreserved PBMC were stained after overnight recovery with antibodies against human CD2, CD3, CD16, CD56, CD57, and NKG2A (extracellular) followed by FcεRIγ and PLZF (intracellular) fluorochrome-conjugated antibodies for flow cytometry. The gating strategy shown proceeds through selection of (**A**) the lymphocyte population, (**B**) exclusion of dead cells, and (**C**) selection of CD3^neg^CD56^pos^ NK cells for further analysis of (**D**–**F**) CD2, CD16, CD57, NKG2A, FcεRIγ, and PLZF expression (**G**) Percent CD2^pos^, (**H)** CD16^pos^, (**I**) CD57^pos^, (**J**) NKG2A^pos^, (**K**) FcεRIγ^neg^, and (**L**) PLZF^neg^ NK cells were compared between the NKG2C^null^ and NKG2C^match^ groups. Data were displayed as mean values with SD and differences compared by the Student’s *t* test when data are normally distributed (**K**,**L**). Data are displayed as median with IQR and differences are compared by the Mann-Whitney test when they were not normally distributed (**G**–**J**). Statistically significant differences are shown on graphs above lines spanning comparison groups.

**Figure 2 viruses-11-00239-f002:**
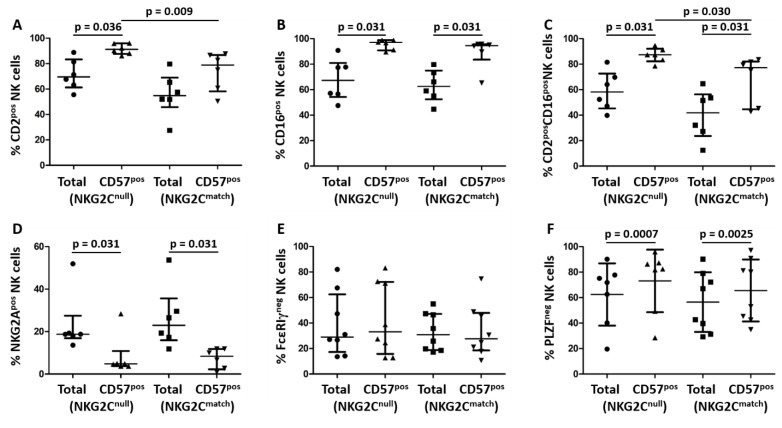
The maturation pattern of NK cells from the NKG2C^null^ and NKG2C^match^ groups. The same gating strategy as described in Figure 1 was used to compare (**A**) CD2, (**B**) CD16, (**C**) CD2/CD16, (**D**) NKG2A, (**E**) FcεRIγ, and (**F**) PLZF expression on the entire NK cell population and CD57^pos^ mature NK cells both within and between groups. The percentage of a particular marker positive or negative within the CD57^pos^ population was calculated as (% marker^pos/neg^ within the CD57^pos^ NK fraction/% total CD57^pos^ NK cells) × 100. Data were displayed as median values with IQR and the probability of differences between groups was calculated by the Mann-Whitney *U* test when data were not normally distributed with paired comparisons within groups performed by the Wilcoxon signed rank test (**A**–**E**). Data were displayed as mean values with SD and differences between groups compared by the Student’s *t* test when data were normally distributed with paired comparisons within groups carried out using the Student’s paired *t* test (**F**). Statistically significant differences are shown on graphs above lines spanning comparison groups.

**Figure 3 viruses-11-00239-f003:**
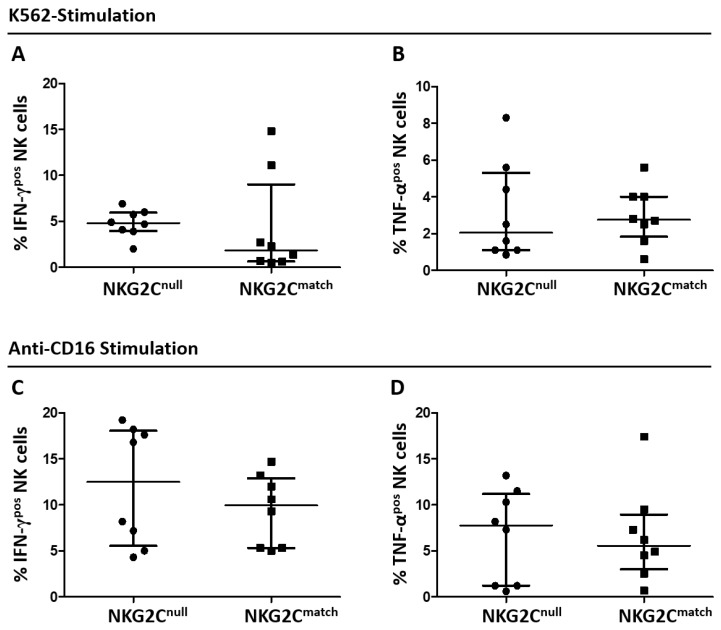
Comparison of cytokine responses to stimulation through NCRs or CD16 for NK cells from NKG2C^null^ and NKG2C^match^ groups. After overnight recovery, cryopreserved PBMC from NKG2C^null^ and NKG2C^match^ individuals were incubated with K562 cells (**A**,**B**) in 1 mL lymphocyte medium at E:T 5:1 (2 × 10^6^ PBMC:4 × 10^5^ K562) or with the 3G8 mAb (**C**,**D**) at 1 µg/mL for 16 h at 37 °C with 5% CO_2_. Brefeldin A was added to a final concentration of 10 μg/mL after the first hour. Post-stimulation, PBMC were labeled with fluorescence-conjugated antibodies to identify NK cells expressing IFN-γ (**A**,**C**) and TNF-α (**B**,**D**). Data were displayed as mean values with SD and differences compared by the Student’s *t* test when data were normally distributed (**B**–**D**) or displayed as median with IQR and differences compared by the Mann-Whitney *U* test when they were not normally distributed (**A**). There was no significant difference between groups in median expression of IFN-γ after K562 stimulation or mean expression of TNF-α among NK cells between groups. Statistically significant differences are shown on the graph above a line spanning the comparison groups.

**Figure 4 viruses-11-00239-f004:**
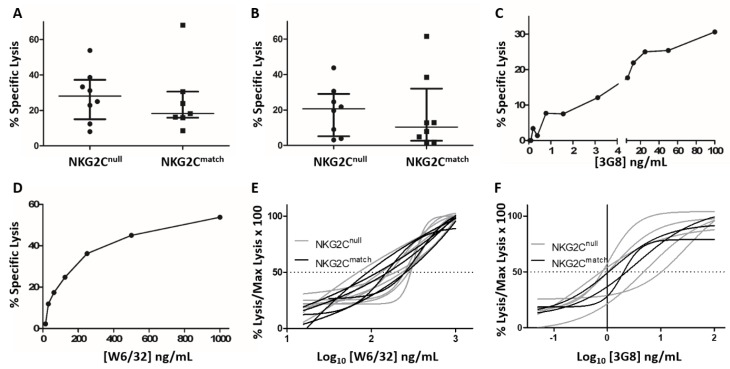
Comparison of cytotoxicity triggered through CD16 for NKG2C^null^ and matched control groups. After overnight recovery, cryopreserved PBMC were incubated with ^51^Cr-labeled (**A**) P815 cells and anti-CD16 (3G8 mAb) or with (**B**) anti-HLA class I-coated C1R-B27 cells at an E:T 50:1. The probability of a significant difference in median lysis between groups in either assay was assessed by the Mann-Whitney *U* test. Representative titrations of PBMC incubated with (**C**) P815 cells and 3G8 mAb at concentrations ranging from 0.05 to 100 ng/mL and (**D**) anti-HLA class I-coated C1R-B27 cells at concentrations ranging from 1.7 to 100 ng/mL in 5 h cytotoxicity assays. EC_50_ values were estimated for each subject from Ab concentrations yielding half-maximal target cell lysis. Titration curves normalized to 100 for maximum observed killing with each subject tested are shown for (**E**) W6/32 and (**F**) 3G8.

**Table 1 viruses-11-00239-t001:** Characteristics of HIV-infected NKG2C^null^ and matched control groups.

	NKG2C^null^	NKG2C^match^	*p* Value
Age in years (mean ± SD)	50.1 ± 9.6	48.3 ± 10.0	ns
Sex	7 ♂, 1 ♀	7 ♂, 1 ♀	ns
CD4^pos^ T cell nadir (mean ± SD) ^a^	295 ± 223	240 ± 208	ns
% NK cells (mean ± SD) ^b^	8.7 ± 4.7	11.2 ± 5.4	ns
Anti-CMV IgG (mean ± SD) ^c^	1.34 ± 0.41	1.10 ± 0.37	ns
% HCMV-specific CD8^pos^ T cells (median with IQR) ^d^	7.4 (2.7–11.4)	1.6 (0.3–4.5)	0.05

^a^ Lowest recorded CD4^pos^ T cell count/μL peripheral blood. ^b^ Percentage of lymphocytes CD56^pos^CD3^neg^. ^c^ Optical density of ELISA reading with plasma diluted 1:500. ^d^ Percentage of CD8^pos^ T cells producing IFN-γ following exposure to overlapping peptides from HCMV pp65 and IE-1 proteins.

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
