# Peer review of "Cytomegalovirus-Driven Adaption of Natural Killer Cells in NKG2Cnull Human Immunodeficiency Virus-Infected Individuals"

_viruses, 2019, doi:10.3390/v11030239_

Round 1
Reviewer 1 Report
- Are HIV patients in this study treated with HAART? if yes, for how long? If no, indicate HIV viral load in Table 1 and discuss influence of active infection on findings.
- inclusion of a group whose NK cells of this subset did differ as would have been expected would have been a good addition to the data set. Perhaps this would have been from an aged population? Something to contrast, or give reference to the data from the HIV-infected group. And how do these data compared to healthy controls, in age-matched settings? This also would be a good data reference.
- is this an issue of interest only in HIV infection? what about other chronic diseases that influence NK cell function (e.g. HCV)? this should be discussed.
- the effect of "time" on NK function is not really addressed with reference to aging. Is this an issue that should be addressed, or what is referred to in citing ref 4? Then, is the phenomenon in HIV infection an expression of early aging of the immune system? please discuss, if relevant.
- ref. for K562 experiments? And are the IFN and TNF data in those experiments meant to infer that the NK cells are able to kill their targets, through this indirect measure? This would be reliant on the fact that NK cells in HIV infection are polyfunctional, but I presume this can be compromised, as is the case in CD8+ T-cells in chronic viral infections. Can the lysis of the K562 cells be monitored in this assay? How can the data from these experiments be reconciled to that which was observed in the subsequent cytotoxicity assays? This was not discussed.
The authors should be commended for the successful matching of their study groups by age, duration of HIV infection and CD4 count: this is not always feasible. Well done!
Also, it is appreciated that the authors demonstrated care in evaluating data for normal distribution, applying appropriate statistical analyses as a result. It is always a challenge to manage the individual variation of responses measured in human primary immune cell culture.
Author Response
Reviewer 1 – responses italicized
- Are HIV patients in this study treated with HAART? if yes, for how long? If no, indicate HIV viral load in Table 1 and discuss influence of active infection on findings.
All of the HIV-infected subjects in this study are treated with cART. The NKG2C nulls identified were followed on treatment in our study for between 5 and 23 years and the HIV-infected subjects we selected as matches were followed on treatment between 4 and 23 years. There were occasional blips or breakthroughs in virus load over this period in both groups, but virus load was mostly maintained below detectable levels with cART. A statement regarding cART treatment of the subjects has been included in the appropriate section of the results on lines 173-177.
- inclusion of a group whose NK cells of this subset did differ as would have been expected would have been a good addition to the data set. Perhaps this would have been from an aged population? Something to contrast, or give reference to the data from the HIV-infected group. And how do these data compared to healthy controls, in age-matched settings? This also would be a good data reference.
A number of previous publications cited in our manuscript have reported that in NKG2C-expressing individuals, NKG2C+ cells that expand due to HCMV infection are more responsive to stimulation through CD16 in terms of cytokine expression. This was also shown in an HIV-infected study group (Peppa, D. et al. Front. Immunol. 2018) with comparison to healthy controls. We have added reference to this study in the introduction to our manuscript on lines 57-59.
- is this an issue of interest only in HIV infection? what about other chronic diseases that influence NK cell function (e.g. HCV)? this should be discussed.
We have added some commentary on this issue to the discussion with reference to a recent publication (Malone, D. F. G. et al. Front. Immunol. 2017) indicating that the CMV-driven expansion of NKG2C+ NK cells is unaffected by chronic HBV, HCV or HDV infection relative to uninfected controls on lines 321-323.
- the effect of "time" on NK function is not really addressed with reference to aging. Is this an issue that should be addressed, or what is referred to in citing ref 4? Then, is the phenomenon in HIV infection an expression of early aging of the immune system? please discuss, if relevant.
Reference 4 refers to the age-related expansion of CMV-specific T cells in CMV-infected individuals. This is accelerated in HIV infection as is the expansion of NKG2C+ NK cells. We and others do consider this an aspect of early aging of the immune system and have added several sentences to the discussion addressing this issue on lines 389-393.
- ref. for K562 experiments? And are the IFN and TNF data in those experiments meant to infer that the NK cells are able to kill their targets, through this indirect measure? This would be reliant on the fact that NK cells in HIV infection are polyfunctional, but I presume this can be compromised, as is the case in CD8+ T-cells in chronic viral infections. Can the lysis of the K562 cells be monitored in this assay? How can the data from these experiments be reconciled to that which was observed in the subsequent cytotoxicity assays? This was not discussed.
In this case we wanted to explicitly investigate cytokine production in response to stimulation through the natural cytotoxicity receptors (NCR). By measuring cytokine production through flow cytometry, we could compare the responses of different NK cells subsets, which is not possible in standard cytotoxicity assays. It could be inferred that the NK cells would also kill the K562s if cytokine responses were triggered through the NCR or if surface CD107a expression increased, but it is not definitive. We focussed on antibody-dependent cellular cytotoxicity in subsequent assays as this is what has been reported to be affected with CMV-driven maturation and differentiation. Although the triggering is through CD16 in this case and not NCR, it does demonstrate that the NK cells are competent to mediate killing. A comment has been added in the discussion lines 341-344.
The authors should be commended for the successful matching of their study groups by age, duration of HIV infection and CD4 count: this is not always feasible. Well done!
Also, it is appreciated that the authors demonstrated care in evaluating data for normal distribution, applying appropriate statistical analyses as a result. It is always a challenge to manage the individual variation of responses measured in human primary immune cell culture.
Thank you for your constructive critique and positive comments.

Reviewer 2 Report
This paper compares Natural Killer cell phenotypic differentiation and function within PBMC from HIV-1 infected NKG2Cnull/null individuals to matched for age, NK cell frequencies, CD4+ T cell counts, HCMV sero-status). The paper is very well written, the experiments are logically conceived and sequenced, accurately executed and presented and the conclusions are appropriate to the data set as presented. The authors conclude that although NK cells from NKG2C-/- HIV+ individuals and HIV- controls are phenotypically similar, CD2+ cells are present at higher frequencies indicating a greater involvement of CD2 in expansions observed in the HIV+ subgroup. The authors also demonstrate similar functional capacity in HIV+ and HIV- NKG2C-/- subgroups.
This study is unique in its investigation of NK cell function NKG2C-/- HIV-+ individuals and merits publication. There are, however some points, which I feel will clarify and extend the presented dataset, if addressed.
1. It is critical to state whether the HIV-1+ subgroup are treatment naïve or receiving anti-retroviral therapy, and information included on duration of treatment, viral loads and nadir CD4+ T cell counts included where relevant. This is important as previous data (Brunetta et al doi: 10.1097/QAD.0b013e3283328d1f- your reference 13) show that the impacts of HIV-1 infection on the NK cell compartment start to reverse after 2 years or ART. Please also include some discussion of your data in the context of anti-retroviral therapy.
2. For Figure 1 please also show gating strategies for CD57, CD16, FcER1g and PLZF flow cytometric analysis.
3. Also in either figure 1 or 2, please show data for the MFI of CD16, CD2 and CD57. This is important since in addition to enhanced frequencies, ‘adaptive NK cells’ have higher levels of CD2, CD16 and CD57. As such, these analyses could reveal additional differences not revealed by simple assessment of cell frequencies.
4. It would be very informative to include some data on cells from NKG2C+/+ or -/- individuals for comparison if this is to hand – these individuals comprise the vast majority of donors studied, particularly in North American and European settings and therefore represent the normal situation
5. Please justify the statement in the abstract ‘Lacking NKG2C diminishes resistance to HIV infection’. Has this actually been fully demonstrated?
6. In the discussion, line 367-370. Please clarify this point – did the authors also compare T cell and NK cell responses according to NKG2C genotype or are they referring to the increase frequency of HCMV-specific CD8+ T cells in HIV-1+ compared to HIV-1- individuals shown in table 1. If the latter, the interpretation may be different - relating more to the peak HCMV viral loads, duration of HCMV infection and frequencies of HCMV reactivations being greater in the HIV-1 infected group.
7. In NKG2C heterozygous or homozygous individuals it is clear that there are HCMV associated expansions of NKG2C+ NK cells, as demonstrated in many studies for both populations of healthy individuals and in transplantation settings. Please qualify the statement in the final paragraph of the discussion ‘Our results imply that NKG2C expression is not heavily involved in driving NK cell differentiation in HCMV infection..…’ to clarify that HCMV associated expansions are observed in NKG2C- NK cells from HCMV+ individuals with or without the gene deletion, indicating alternative pathways of HCMV mediated expansions (CD2/CD16/ short form KIR).
Author Response
Reviewer 2 - responses italicized
This paper compares Natural Killer cell phenotypic differentiation and function within PBMC from HIV-1 infected NKG2Cnull/null individuals to matched for age, NK cell frequencies, CD4+ T cell counts, HCMV sero-status). The paper is very well written, the experiments are logically conceived and sequenced, accurately executed and presented and the conclusions are appropriate to the data set as presented. The authors conclude that although NK cells from NKG2C-/- HIV+ individuals and HIV- controls are phenotypically similar, CD2+ cells are present at higher frequencies indicating a greater involvement of CD2 in expansions observed in the HIV+ subgroup. The authors also demonstrate similar functional capacity in HIV+ and HIV- NKG2C-/- subgroups.
This study is unique in its investigation of NK cell function NKG2C-/- HIV-+ individuals and merits publication. There are, however some points, which I feel will clarify and extend the presented dataset, if addressed.
1. It is critical to state whether the HIV-1+ subgroup are treatment naïve or receiving anti-retroviral therapy, and information included on duration of treatment, viral loads and nadir CD4+ T cell counts included where relevant. This is important as previous data (Brunetta et al doi: 10.1097/QAD.0b013e3283328d1f- your reference 13) show that the impacts of HIV-1 infection on the NK cell compartment start to reverse after 2 years or ART. Please also include some discussion of your data in the context of anti-retroviral therapy.
In response to a similar question from reviewer 1, we added a statement on treatment and duration of treatment for the 2 groups on lines173-177. All the subjects in our study (NKG2Cnull and matches) are HIV-infected. While the Brunetta paper reported some reversal of the effects of HIV on the NK cell compartment (specifically NKG2C/NKG2A ratio), this may have been with treatment early in infection. Several other studies (refs. 10-12 and 22) and our own experience indicate that changes to the NK cell population mediated in response to CMV infection in terms of expansion of NKG2C-expressing cells are stable, both in HIV-infected individuals and others.
2. For Figure 1 please also show gating strategies for CD57, CD16, FcER1g and PLZF flow cytometric analysis.
Gating strategies for CD57, CD16, FceR1g and PLZF are now shown in Figure 1.
3. Also in either figure 1 or 2, please show data for the MFI of CD16, CD2 and CD57. This is important since in addition to enhanced frequencies, ‘adaptive NK cells’ have higher levels of CD2, CD16 and CD57. As such, these analyses could reveal additional differences not revealed by simple assessment of cell frequencies.
We agree that a quantitative analysis of expression levels on a per cell basis could be meaningful. Unfortunately, the flow cytometry was not done at the same time or calibrated with fluorescent beads beyond daily calibration of the flow cytometer, so MFIs are not directly comparable.
4. It would be very informative to include some data on cells from NKG2C+/+ or -/- individuals for comparison if this is to hand – these individuals comprise the vast majority of donors studied, particularly in North American and European settings and therefore represent the normal situation
Surprisingly, we identified 3 NKG2Cnull healthy control subjects, all of whom were HCMV-seronegative. We would be unable to conduct a meaningful comparison with data or subjects on hand.
5. Please justify the statement in the abstract ‘Lacking NKG2C diminishes resistance to HIV infection’. Has this actually been fully demonstrated?
Reference 25 describes a study of 433 HIV-infected individuals and 280 controls in which they found that lacking the gene for NKG2C was associated with a greater risk for HIV infection, higher pre-treatment virus loads and faster disease progression. References 26 and 27 suggest a direct role for NKG2C+ NK cells in controlling HIV replication.
6. In the discussion, line 367-370. Please clarify this point – did the authors also compare T cell and NK cell responses according to NKG2C genotype or are they referring to the increase frequency of HCMV-specific CD8+ T cells in HIV-1+ compared to HIV-1- individuals shown in table 1. If the latter, the interpretation may be different - relating more to the peak HCMV viral loads, duration of HCMV infection and frequencies of HCMV reactivations being greater in the HIV-1 infected group.
Table 1 compares the frequency of HCMV-specific CD8+ T cells in the two HIV-1+ groups, HIV+ NKG2C nulls and HIV+ matched controls.
7. In NKG2C heterozygous or homozygous individuals it is clear that there are HCMV associated expansions of NKG2C+ NK cells, as demonstrated in many studies for both populations of healthy individuals and in transplantation settings. Please qualify the statement in the final paragraph of the discussion ‘Our results imply that NKG2C expression is not heavily involved in driving NK cell differentiation in HCMV infection..…’ to clarify that HCMV associated expansions are observed in NKG2C- NK cells from HCMV+ individuals with or without the gene deletion, indicating alternative pathways of HCMV mediated expansions (CD2/CD16/ short form KIR).
This statement has been qualified as indicated in the discussion lines 399-402.